# Swift Sampler: Efficient Learning of Sampler by 10 Parameters

**Jiawei Yao**[1*]  **Chuming Li**[2,3*]  **Canran Xiao**[4†]
[1] University of Washington [2] Shanghai Artifcial Intelligence Laboratory
[3] The University of Sydney [4] Central South University
jwyao@uw.edu, chli3951@uni.sydney.edu.au, xiaocanran@csu.edu.cn

## Abstract

Data selection is essential for training deep learning models. An effective data sampler assigns proper sampling probability for training data and helps the model converge to a good local minimum with high performance. Previous studies in data sampling are mainly based on heuristic rules or learning through a huge amount of time-consuming trials. In this paper, we propose an automatic **swift sampler** search algorithm, **SS**, to explore automatically learning effective samplers efficiently. In particular, **SS** utilizes a novel formulation to map a sampler to a low dimension of hyper-parameters and uses an approximated local minimum to quickly examine the quality of a sampler. Benefiting from its low computational expense, **SS** can be applied on large-scale data sets with high efficiency. Comprehensive experiments on various tasks demonstrate that **SS** powered sampling can achieve obvious improvements (e.g., 1.5% on ImageNet) and transfer among different neural networks. Project page: https://github.com/Alexander-Yao/Swift-Sampler.

## 1  Introduction

Training data plays a pivotal role in deep learning tasks. The sampling probability of data in the training process can significantly influence the performance of the learned model. A set of works [Jiang et al., 2019, Katharopoulos and Fleuret, 2018, Needell et al., 2014, Johnson and Guestrin, 2018, Han et al., 2018, Hu et al., 2024] have demonstrated improvements in model training by sampling data according to different features and under different rules. These studies reveal that how to sample examples during training is nontrivial and sampling data in a totally uniform way is not always the optimal choice.

Previous works on data sampling strategy include two main categories: human-defined rules and learning-based methods. Some works [Han et al., 2018, Jiang et al., 2019, Katharopoulos and Fleuret, 2018, Hacohen and Weinshall, 2019, Kumar et al., 2010] proposed manually designed rules, such as setting the sampling probability of training examples with loss values larger than a threshold as zero, or making the sampling probability proportional to the gradient norms. Such human-defined rules are designed for specific tasks and hardly adapts to different scenarios, because optimal sampling strategy varies among different tasks and data sets. Learning-based methods [Fan et al., 2017, Jiang et al., 2017, Ren et al., 2018] explore an automatic way to assign sampling probability to a given learning task, including sample-based and differential based methods. Sample-based methods take advantage of deep reinforcement learning (DRL) [Fan et al., 2017] to model the training process of the target model as an environment and use a DRL model to learn the optimal sampling strategy during training. These methods need hundreds of replays of the training process and require too much searching cost to be applied on large-scale data sets, e.g., ImageNet [Russakovsky et al., 2015]. Differential based methods [Ren et al., 2018, Shu et al., 2019] assume a completely clean meta dataset and use the inner

---

[*]These authors contribute equally to this work.
[†]Corresponding author.

38th Conference on Neural Information Processing Systems (NeurIPS 2024).

product between the gradients of training data and meta data to sample or reweight training data. The problem is that the clean meta data is not guaranteed to be available for all scenarios, and complex changes are made in the training process.

To explore an automatic way of sampler search, we focus on improving sample-based search methods as they do not require extra meta data and impose no change in the training process when applying the obtained sampler. Typically, a sample-based method repeatedly uses an agent (e.g., Bayesian Optimization or Deep Reinforcement Learning) to sample a sampler and evaluate the **objective function** value of the sampled sampler to update the agent. There are three challenges in designing a sample-based method for automatic sampler search: (1) *High dimension.* A sampler is defined by a vector of sampling probabilities of all instances in the training set. Therefore, the complexity of searching the optimal sampler in a sample-based way increases exponentially with the number of training instances. (2) *Sharpness.* As great difference exists among gradients of various instances in the training set, sampling probability of some instances has a significant impact on the performance of resulted models. Such property results in the sharpness of the objective function for samplers and harms the efficiency of the agent's learning. (3) *Costly evaluation.* The agent seeks a sampler that achieves high performance when used to train the model from scratch. Thus the evaluation of a single sampler causes high computational expense (e.g., training 100 epochs from scratch on ImageNet).

In this paper, we aim at addressing the three problems listed above. For the high dimension problem (1), we define **sampler** as a function mapping the feature of the training data to their sampling probabilities and formulate the search space of sampler as a family of composite functions, which are represented by a small number of hyper-parameters. The amount is much less than the number of training data. Moreover, our formulation of the sampler has a flexible expression covering any features used in previous works and is adaptive to a range of data sets. We carefully choose the features and mapping functions so that the number of parameters to be learned for the sampler is only 10. To relieve the sharpness problem (2) of the objective function, we modify the objective function by designing a transform function to smooth the landscape of the objective function. This transform function balances the amounts of gradient norm in different areas of the definition domain of the objective function. For the costly evaluation problem (3), we use a fast approximation method for learning the network, which is much less expensive than training from scratch. Integrating the designs above, we propose an automatic **swift sampler** learning (**SS**) method, where we choose naive Bayesian Optimization (BO) [Brochu et al., 2010, Klein et al., 2016, Snoek et al., 2012] as the agent. The **swift** here means our formulation of the sampler has a very small volume of parameters, and our method has a high speed of sampler learning, like a swift, the small and fast bird.

We apply **SS** to training neural networks with various sizes, including ResNet-18 and SE-ResNext-101, with training data from different data sets including ImageNet [Russakovsky et al., 2015], CIFAR10 and CIFAR100 [Krizhevsky et al., 2009]. Experiments demonstrate obvious improvements in the performance compared with the baseline and other methods, e.g., 1.5% on ImageNet. During the search process, **SS** optimizes much faster compared with previous automatic methods, and the learned sampler consists of only **10** hyper-parameters. Further analysis shows that the sampler found by SS transfers well among neural networks with various architectures and sizes. The contributions of this paper are summarized as follows:

- We formulate a search space of sampler consisting of a new family of functions. The family of functions is decided by a small number of parameters and also has flexible expression. The reduced dimension facilitates the application of **SS** on large-scale data sets.

- To improve the efficiency of the optimization of the sampler, we smooth the sharp objective function of the sampler search problem with a carefully designed transform function.

- We use an approximation method to approximate the local minima of the sampler efficiently tried by the agent. This method makes **SS** fast and capable of improving the performance of models on large data sets.

## 2 Related Work

**Hard-wired Methods** Hard-wired methods have fixed sampling rules and focus on a few particular problems, e.g., imbalance, noise and training speed up. Each problem needs respective hand-crafted rules and the designs are based on specific understandings of data. Thus, these methods hardly generalizes over a wide range of data sets. In [Han et al., 2018], the sampling method focuses on denoising problem, and neglects instances with loss values larger than a gradually increasing

threshold. It needs prior knowledge on the ratio of noisy instances in the data set. For imbalance, [He and Garcia, 2009, Maciejewski and Stefanowski, 2011, Lin et al., 2017, Wang et al., Li et al., 2022] propose methods to boost the training of imbalanced data by optimizing the sampling or weighting (a 'soft' sampling way) of instances, and develop different features as the signals of imbalance. Importance sampling [Hacohen and Weinshall, 2019, Kumar et al., 2010, Ma et al., 2017, Graves et al., 2017] speeds up the training process by giving larger sampling probabilities to instances with higher loss values. But it does not fit noisy data sets. Curriculum learning (CL) methods [Hacohen and Weinshall, 2019, Kumar et al., 2010, Jiang et al., 2015] are also related to our work. It shows that some particular sampling orders benefit training process.

**Learning-based Methods** Like recently proposed automated loss function search [Li et al., 2019a] and augmentation policy search methods [Cubuk et al., 2018, Lin et al., 2019, Tian et al., 2020], many recent works also explore automatically learning how to optimize data sampling. They can achieve generalization on various scenarios. [Fan et al., 2017] proposes a RL-based framework to optimize the data sampling in different training stages under the setting of mini-batch SGD. It models the RL agent as the teacher to guide the training of the student model. This method requires multiple runs of training from scratch, thus does not fit large data sets such as ImageNet. [Li et al., 2019b] proposes a similar RL-based framework to reweight training instances, which can be viewed as a soft sampling method. Some other related works [Ren et al., 2018, Shu et al., 2019] propose differential based methods. It makes use of the gradients of loss of a completely clear meta dataset, sampling or weighting the training datum via the inner product between their gradients and meta data gradients. In summary, the methods in [Fan et al., 2017, Li et al., 2019b] require lots of parameters with high computational costs and needs extra data [Ren et al., 2018, Jiang et al., 2017, Shu et al., 2019]. In comparison, our approach is fast, requires few parameters but no extra data.

**Bayesian Optimization** Bayesian Optimization (BO) has shown its great potentials in optimizing the hyper-parameters of learning tasks. The method in [Snoek et al., 2012] first introduces BO to optimize the hyper-parameters of various tasks, including SVM and regression. FABOLAS [Klein et al., 2016] constructs a BO-based method to simultaneously optimizes computation cost and information gain w.r.t hyper-parameters of SVM and training data size. It is at most 100 times faster than previous methods. Recently, NASBOT [Kandasamy et al., 2018] explores the usage of BO in neural architecture search by mathematically defining the distance between two architectures. The distance enables the design of kernel for BO.

In our approach, the use of Bayesian Optimization is not our contribution, although naive Bayesian Optimization is used for solving our problem. Therefore, the recent advances in Bayesian Optimization can be also used for our approach.

## 3 Method

We introduce **SS** in this section. First, we describe the problem of sampler search in a bilevel optimization framework, consisting of the outer loop and the inner loop. The **outer loop** uses an agent to learn to sample the sampling probabilities of training instances and optimizes the performance of the target model when trained under the sampled sampling probabilities. And the **inner loop** minimizes the loss of model parameters on the training set, with sampling probabilities given by the outer loop. Further, we formulate the sampler, define its search space, and design a transform function to smooth the curvature of the objective function of the outer loop. Then, we introduce our agent for the learning of sampling in the outer loop. Finally, we propose a highly efficient method to approximate the local minima of given sampling probabilities rather than training from scratch.

### 3.1 Problem Formulation

A common practice of training deep neural networks (DNN) is using mini-batch SGD to update the network parameters. The batches are formed by uniformly sampling from the training set. Sampling methods usually take different settings where the sampling probability of training instances are optimized [Hacohen and Weinshall, 2019, Kumar et al., 2010, Jiang et al., 2015, Ma et al., 2017, Jiang et al., 2019, Katharopoulos and Fleuret, 2018, Needell et al., 2014, Johnson and Guestrin, 2018]. In this work, we focus on finding a static sampling probability which guides the parameters of target DNN to the local optimum with the best performance on the validation set. We formulate this problem as follows.

For a target task, e.g., image classification, its training set and validation set are respectively denoted by $D_t$ and $D_v$, and the parameters of the target model are denote by $\boldsymbol{w}$. Each sample $x_i$ for $x_i \in D_t$ has its corresponding sample probability $\tau(x_i)$. We define the probability function $\tau$ as **sampler**. We formulate the optimization of $\tau$ as a bilevel problem.

**The inner loop** learns the network parameters that minimize the expected loss on the target task under the sampling probability $\tau$ given by the outer loop. Denoted by $\boldsymbol{w}^*(\tau)$ the local minima of network parameters trained with loss $L(x; \boldsymbol{w})$ and sample probability $\tau$, $\boldsymbol{w}^*(\tau)$ is obtained as follows:

$$\boldsymbol{w}^*(\tau) = \arg\min E_{x \sim \tau}[L(x; \boldsymbol{w})]. \tag{1}$$

**The outer loop** uses an agent to search for the best sampler $\tau$. Specifically, the network with parameters $\boldsymbol{w}^*(\tau)$ obtained from the inner loop is used for searching the sampler $\tau$ that has the best score $P(D_v; \boldsymbol{w}^*(\tau))$ on validation set $D_v$, where $P(D; \boldsymbol{w})$ is the performance score of parameters $\boldsymbol{w}$ on a given data set $D$ and $P(D_v; \boldsymbol{w}^*(\tau))$ is our objective function. The outer loop problem is described as:

$$\tau^* = \arg\max_\tau P(D_v; \boldsymbol{w}^*(\tau)). \tag{2}$$

Both the outer loop and the inner loop are difficult to solve. The outer loop is a high-dimension optimization problem, where the optimized sampler $\tau(x_i)$ has a dimension equal to the number of training data. In addition, the objective function $P(D_v; \boldsymbol{w}^*(\tau))$ for the agent of the outer loop encounters a sharpness problem. Further, given a $\tau$ from the outer loop, the inner loop needs to train from scratch with $\tau$ to get the local minima $\boldsymbol{w}^*(\tau)$, which involves high computational cost. We introduce our solutions to the three problems hereinafter.

### 3.2 Sampler Formulation

The complexity of the optimization problem increases exponentially with the dimension of $\tau$, which is the number of training data. However, modern hyper-parameter optimization methods like BO can hardly handle more than 30 hyper-parameters [Brochu et al., 2010]. Hence we reduce the dimension of $\tau$ by a simple formulation. The high dimension comes from independence among training samples. However, we assume that the optimal sampler lies in a more compact subdomain. Instead, we restrict that the difference in the sampling probabilities of two training instances is bounded by their distance in the feature space, e.g., the (loss, entropy) space.

**Example** As an intuitive example, we consider cropped images in ImageNet [Russakovsky et al., 2015] dataset as samples and define a one-dimension feature space where the feature is the loss value. Cropped images with the highest loss values have **small distance** between each other in this feature space. Further, as we show in Sec. 4, most of them are noisy instances, which should all be assigned sampling probabilities near 0. It means they also have **small sampling probabilities difference**.

Mathematically, the intuition above can be formulated as constraining samplers to satisfy Lipschitz condition:

$$|\tau(x_i) - \tau(x_j)| \leq C \cdot \|\boldsymbol{f}(x_i) - \boldsymbol{f}(x_j)\|_2, \tag{3}$$

where $\|\boldsymbol{f}(x_i) - \boldsymbol{f}(x_j)\|_2$ is the $L_2$ distance in the space of feature vector $\boldsymbol{f}$ and $C$ is a real positive number.

**Dimension Reduction** With the above consideration, we define $\tau(x)$ as a multivariate continuous function of the features of instance $x$, described by

$$\tau(x) = F(\boldsymbol{f}_1(x), \boldsymbol{f}_2(x), ... \boldsymbol{f}_N(x)), \tag{4}$$

where $F$ is a multivariate continuous function and $f_i(x)$ is $i$-th feature of example $x$ ($i = 1, ..., N$). The choices of features have been explored in a wide range, e.g., loss in [He and Garcia, 2009, Maciejewski and Stefanowski, 2011, Jiang et al., 2019, Katharopoulos and Fleuret, 2018], and density in [Fan et al., 2017]. Our choice of features is discussed in the latter sections.

We consider a family of $F$ which have flexible expression and low dimension. $F$ are formed by: (1) a univariate piecewise linear function $H$ defined on the period $[0, 1]$, (2) a univariate transform function $T$ which balances the density of the gradient of instances along the period $[0, 1]$ to smooth the objective function of the agent and (3) an multivariate aggregation function $G(x)$ which aggregates

information from all features $f$:

$$F\left(\boldsymbol{f}_1\left(x\right), \boldsymbol{f}_2\left(x\right), ..., \boldsymbol{f}_N\left(x\right)\right) = H\left(T\left(G\left(x\right)\right)\right), \tag{5}$$

$$G\left(x\right) = \sum_{i=1}^{N} \boldsymbol{c}_i \cdot \boldsymbol{f}_i\left(x\right), \tag{6}$$

where $\boldsymbol{c}_i$ are real-value coefficients aggregating features and $T$ maps the input $G\left(x\right)$ to a value in the closed interval $[0, 1]$. The definition of $T$ and the explanation of why it smooths the objective function of the agent are given in the latter part of this section. To aggregate different features in a unified scale $[0, 1]$, we use cumulative distribution function $cdf$ of the features:

$$\boldsymbol{f}_i\left(x\right) = cdf\left(\boldsymbol{f}_i^o\left(x\right)\right), \tag{7}$$

where $\boldsymbol{f}_i^o$ denotes the original numerical value of features, e.g., cross entropy loss in classification with range $[0, \infty)$.

The $G\left(x\right)$ in Eqn. 6 takes a linear aggregation form and projects the whole feature space to a single dimension, which distinguishes the importance of different instances to the greatest extent. $H$ is a continuous piecewise linear function because it can fit a wide range of continuous function when the number of segments is large enough.

With such definition, $F$ is decided by parameters in functions $H, T$, and $G$, much smaller than the number of the training data.

$\tau$ defined by Eqn. 4, Eqn. 5, Eqn. 6, and Eqn. 7 satisfies Lipschitz condition when $cdf$ and transform $T$ are continuous in close definition domains.

**Search Space** For the univariate piecewise linear function $H$, we define $\boldsymbol{e}_0, \boldsymbol{e}_1, ..., \boldsymbol{e}_S$ as the positions of endpoints, and define $\boldsymbol{v}_0, \boldsymbol{v}_2, ..., \boldsymbol{v}_S$ as the values on the endpoints. $\boldsymbol{e}_0$ and $\boldsymbol{e}_S$ are fixed as 0 and 1. Then the feasible domain (search space) of the sampler $\tau$ is:

$$\boldsymbol{e}_s \leq \boldsymbol{e}_j, \boldsymbol{v}_s \in [0, 1], \forall\, 0 \leq s \leq j \leq S, \tag{8}$$

$$\boldsymbol{c}_i \in [-1, 1], \forall\, 1 \leq i \leq N. \tag{9}$$

In our implementation, $S$ is 4 and $N$ is 2, resulting in totally only 10 hyper-parameters.

**Smooth the Objective Function** When the curvature of objective function $P\left(D_v; \boldsymbol{w}^*\left(\tau\right)\right)$ is too large, i.e., with a sharp landscape, agents like BO would need more trials to find the maxima [Brochu et al., 2010]. Such sharpness problem exists in our scenario. This sharpness problem is caused by the imbalance between the gradients of training instances lying in different segments of the piecewise linear function $H$. For the function $H$ whose input is $t(x) = T(G(x))$ and one of its segment $[\boldsymbol{e}_{i-1}, \boldsymbol{e}_i]$, if the instances $x$ with $t(x)$ lying the interval $[\boldsymbol{e}_{i-1}, \boldsymbol{e}_i]$ have much larger gradients than the other intervals, then a little variation of the value $\boldsymbol{v}_i$ or $\boldsymbol{v}_{i-1}$ will cause a large difference on the overall gradient from the whole training data, which results in large difference in the objective function $P$. For example, the worst trained 10% instances of ImageNet contains over 90% gradients norm of the whole set. It means the curvature of objective function $P$ around $\tau^*$ is likely to be sharp and the value of $P$ is only distinguishable in a little sub-domain of $\tau$. If the maxima $\tau^*$ has such sharpness problem, the efficiency of the agent would be reduced as we show in Sec. 4.4. To smooth the objective function, we define the cumulative gradient function $cgf$:

$$T(u) = cgf\left(u\right) = \frac{\sum_{x_i \in D_t, G(x_i) <= u} grad\left(x_i\right)}{\sum_{x_i \in D_t} grad\left(x_i\right)}, \tag{10}$$

where $grad\left(x_i\right)$ is the gradient norm of $x_i$. It can be easily proved that any two intervals in the domain of $H$ with equal lengths contain the same amount of total gradient norm, as $cgf\left(u\right)$ itself is the cumulative gradient norm. This design brings the search on sampler optimization problem with high efficiency. We conduct experiment in Sec. 4.4 to verify the necessity of objective function smoothing via comparing $T = cgf$ and $T = cdf$.

**Static vs. Varying Features** In our formulation, $\tau\left(x\right)$ is a static function, which restricts that the features do not change in the training process as well. This means a pre-trained model should be used to produce features for $D_t$ rather than the model currently being trained. The reason is that a model being trained often forgets and re-memorizes part of $D_t$ [Toneva et al., 2018], resulting in jitters and noises for the features like loss or entropy. Thus we empirically find that fixed features are more effective than varying features for learning samplers.

### 3.3 Optimization

**Bayesian Optimization** For simplicity, we define $z = [e_0, ..., e_S, v_0, ..., v_S, c_1, ..., c_N]$, and replace $P(D_v; w^*(\tau))$ with $P(z)$ in the latter discussion because $\tau$ and $z$ are one-to-one mapped. In the outer loop, to find maxima $z^*$ of $P$ with as few trials as possible, we explore the advantage of Bayesian Optimization [Brochu et al., 2010]. Bayesian Optimization is an approach to optimize objective functions that take a long time (e.g., hours) to evaluate. It is best-suited for optimization over continuous domains with the dimension less than 30, and tolerates stochastic noise in function evaluations, which are exactly characteristics of our problem. Recent works proved the potential of Bayesian Optimization (BO) on hyper-parameter tuning of DNN [Klein et al., 2016, Snoek et al., 2012]. Given the black-box performance function $P : \mathcal{Z} \to \mathcal{R}$, BO aims to find an input $z^* = \arg\max_{z \in \mathcal{Z}} P(z)$ that globally maximizes $P(z)$.

BO requires a prior $p(P)$ over the performance function, and an acquisition function $a_p : \mathcal{Z} \to \mathcal{R}$ quantifying the **utility** of an evaluation at any $z$, depending on the prior $p$. With these ingredients, the following three steps are iterated [Brochu et al., 2010]: (1) find the most promising $z_{t+1} \in \arg\max a_p(z)$ by numerical optimization; (2) evaluate the expensive and often noisy function $Q_{t+1} \sim P(z_{t+1}) + \mathcal{N}(0, \sigma^2)$ and add the resulting data point $(z_{t+1}, Q_{t+1})$ to the set of observations $O_t = (z_j, Q_j), j = 1, ..., t$; (3) update the prior $p(P|O_{t+1})$ and acquisition function $a_{p(P|O_{t+1})}$ with new observation set $O_{t+1}$. Typically, evaluations of the acquisition function $a$ are cheap compared to evaluations of $P$ such that the optimization effort is negligible.

**Gaussian Processes** Gaussian processes (GP) are a prominent choice for $p(P)$, thanks to their descriptive power and analytic tractability [Brochu et al., 2010]. Formally, a GP is a collection of random variables, such that every finite subset of them follows a multivariate normal distribution. To detail the distribution, a GP is identified by a mean function $m$ (often set to $m(z) = 0$), and a positive definite covariance function (kernel) $k$ (often set to RBF kernel [Brochu et al., 2010]). Given history observation $O_t$, the posterior $p(P|O_t)$ follows normal distribution according to the above definition, with mean and covariance functions of tractable, analytic form. It means we can estimate the mean and variance of $P$ on a new point $z_{t+1}$ by marginalize over $p(P|O_t)$. The mean and the variance denote the **expected performance** of and the **potential** of $z_{t+1}$.

**Acquisition Function** The role of the acquisition function is to trade off **expected performance** and **potential** by choosing next tried point $z_{t+1}$. Popular choices include Expected Improvement (EI), Upper Confidence Bound (UCB), and Entropy Search (ES) [Brochu et al., 2010].

In our method, following the popular settings in [Brochu et al., 2010], we choose GP with RBF kernel [Brochu et al., 2010] and a constant $m$ function whose value is the mean of performance $P(z_t)$ of all tried samples. For the choice of acquisition function, we use UCB in all our experiments.

---

**Algorithm 1** SS

---

1: **Inputs:** $E_o$ (BO steps), $E_f$ (fine-tune epochs), $D_t$, $D_v$, $\boldsymbol{f}$, $\boldsymbol{w}_{share}$, $BO$ (the agent)
2: CandidateSamplers $= \emptyset$
3: Initialize($BO$)
4: **for** $s = 1 : E_o$ **do**
5:     $\tau = BO$.sample()
6:     **for** $e = 1 : E_f$ **do**
7:         $\boldsymbol{w}$ = TrainForOneEpoch($\boldsymbol{w}_{share}$,$\tau$,$\boldsymbol{f}$,$D_t$)
8:     **end for**
9:     $P(\tau)$ = P($\boldsymbol{w}$,$D_v$)
10:     $BO$.Update($\tau$,$P(\tau)$)
11:     CandidateSamplers = CandidateSamplers $\cup \{\tau\}$
12: **end for**
13: **Outputs:** CandidateSamplers.Top

---

### 3.4 Local Minima Approximation

The critical problem of the inner loop is how to get $w^*(\tau)$ at an acceptable computational cost. Training from scratch is too computationally expensive to be used on a large data set. Hence, we design a method to obtain an approximation of $w^*(\tau)$ at a limited cost.

Table 1: **SS** results on CIFAR10 and CIFAR100 comparisons with other methods. The number pair X / Y means the Top-1 accuracy on CIFAR10 is X% and on CIFAR100 is Y%.

| Methods | Noise Rate | | | | |
| | 0 | 0.1 | 0.2 | 0.3 | 0.4 |
|---|---|---|---|---|---|
| Baseline | 93.3 / 74.3 | 88.9 / 67.5 | 81.8 / 60.7 | 74.7 / 53.1 | 64.9 / 45.1 |
| REED | 93.4 / 74.3 | 89.4 / 68.1 | 83.4 / 62.5 | 77.4 / 55.2 | 68.6 / 50.7 |
| MN | 93.4 / 74.2 | 90.3 / 69.0 | 86.1 / 65.1 | 83.6 / 59.6 | 76.6 / 56.9 |
| LR | 93.2 / 74.2 | 91.0 / 70.1 | 89.2 / 68.3 | 88.5 / 63.1 | 86.9 / 61.3 |
| **SS** | **93.8 / 75.2** | **91.7 / 71.2** | **90.4 / 69.2** | **89.7 / 64.4** | **88.0 / 62.3** |

First, we focus on the main properties of $w^*(\tau)$: (1) minimizing the expected loss on the train set with sampler $\tau$, (2) undergoing a complete training process. We assume that weight vectors $w$ with the two properties are accurate enough to approximate $w^*(\tau)$ so that we can use this approximated $w^*(\tau)$ to learn $\tau$. Our method initializes the parameters $w$ for different samplers $\tau$ with the same parameters $w_{share}$ which is learned from a complete training process to meet property (2). After this special initialization, we fine-tune $w_{share}$ with the given sampler. For property (1), we refer to recent works on the memorization of DNN [Toneva et al., 2018, Kirkpatrick et al., 2017], which shows DNN tends to fit the data currently used for training and forget historical training instances. As an analogy, our experiments show that the parameters fine-tuned from $w_{share}$ under $\tau$ converge to the same loss level as $w^*(\tau)$ trained from scratch and with enough iterations.

The shared starting point $w_{share}$ is the weight trained from scratch with uniform sampling. Experiments in Sec. 4.4 demonstrate the effectiveness of our approximation method.

The complete process of **SS** is in Algorithm 1. We use BO to explore $E_o$ samplers. In each step, BO agent samples the candidate sampler $\tau$. We use the candidate sampler $\tau$ to produce probabilities with the pre-trained feature $f$ for sampling training data. The sampled training data are then used for fine-tuning the network weight. After the fine-tuning for $E_f$ epochs, BO agent is updated with the candidate sampler $\tau$ and its performance on fine-tuned parameters, then it produces more accurate estimations of the better sampler to facilitate the next sampling.

## 4 Experiment

In this section, we choose three classification and face recognition tasks as benchmarks to illustrate the effectiveness of **SS**: CIFAR10, CIFAR100, ImageNet, and MS1M, including data sets with both small and large sizes. The ablation experiments are also included in Appendix 4.4

In all experiments, the optimization step $E_o$ is fixed as 40, and the fine-tune epochs $E_f$ are set to 5. We set the number of segments $S$ as 4 in all cases and utilize 8 NVIDIA A100 GPUs to ensure efficient processing. We tuned these hyper-parameters by separately increasing them until negligible improvements are obtained on ImageNet. The features and shared start points $w_{share}$ are from the pre-trained models with the same architectures as the target models. We norm the sum $\tau$ to 1 when use it to sample. For the choice of features, we consider the following two features: (1) Loss: Training loss is frequently utilized in curriculum learning, hard example mining and self-paced method. It is a practicable discriptor and is usually viewed as signals of hardness or noise. For our classification benchmarks, we use the Cross Entropy (CE) loss. (2) Renormed Entropy: The entropy of predicted probabilities is also widely used in current methods. To decouple it from CE loss, we delete the probability of target label from the probability vector, and renorm the rest ones in the vector to 1. Then we use the resulted renormed vector to calculate the entropy:

$$E^r(x_i) = -\sum_{j \neq y_i} \frac{p_j}{\sum_{j \neq y_i} p_j} log\left(\frac{p_j}{\sum_{j \neq y_i} p_j}\right), \tag{11}$$

where $p_j$ denotes the predicted probability of the j-th class. A small $E^r$ results from a peak in the distribution over the rest vector, which often implies a misclassification, while a large $E^r$ implies a hard instance.

### 4.1 CIFAR Experiment

CIFAR10 and CIFAR100 are two well-known classification tasks. We explore the performance of our method with the target model ResNet18 on both the original data sets and the data sets with four ratios of noisy labels, 10%, 20%, 30%, and 40%. When generating noisy labels, we random sample instances in each class with the given ratios and uniformly change their labels into the rest incorrect

Table 2: Comparison of Top-1/5 accuracy of **SS** and baseline on ImageNet ILSVRC12. "MB" ,"RN" and "SRN" means MobileNet ResNet and SE-ResNext. **SS(self)**, **SS(R18)** and **SS(R50)** means the the sampler is searched on the target model, ResNet-18 and ResNet-50. All results are averaged over 5 runs, and the deviations are omitted because they are all less than 0.10. It is observed that SS has consistent improvements on Top-1 Acc on all cases, and the performance gain on Top-5 is relatively less because we only use Top-1 Acc as the objective of sampler search.

| Model | MB-v2 | RN-18 | RN-50 | RN-101 | SRN-50 | SRN-101 | Swin-T | Swin-B |
|---|---|---|---|---|---|---|---|---|
| Baseline | 70.4 / 89.7 | 70.2 / 89.4 | 76.3 / **93.1** | 78.0 / 93.8 | 78.6 / 93.9 | 78.9 / 94.4 | 81.2 / 95.5 | 83.5 / 96.5 |
| **SS(self)** | **71.9 / 90.1** | **71.6 / 89.8** | **77.7** / 93.0 | **79.3 / 94.0** | **79.8 / 94.2** | **80.0 / 94.6** | **82.1 / 95.6** | **84.3 / 96.7** |
| **SS(R18)** | - | 71.6 / 89.8 | - | - | 79.5 / 94.2 | 79.8 / 94.5 | - | - |
| **SS(R50)** | - | - | 77.7 / 93.0 | - | 79.6 / 94.2 | 79.8 / 94.5 | - | - |

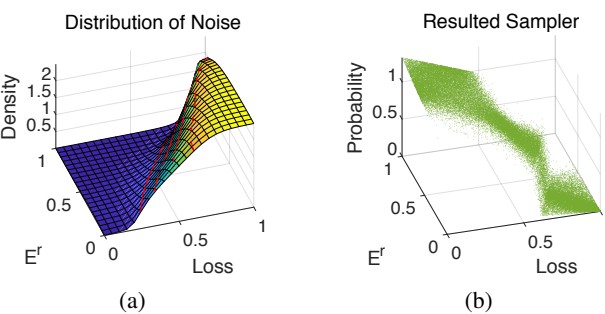

(a)                                          (b)

Figure 1: **A demonstration of the effectiveness of our SS.** (a) The density of noisy instances of noise 40% on CIFAR10 in (Loss,$E^r$) space. (b) The sampling probability of sampler from **SS**. (a)(b) show that **SS** accurately distinguishes the noisy instances and discards them.

classes. We set batch size as 128 and the L2 regularization as 1e-3. The training process lasts 80 epochs, and the learning rate is initialized as 0.1 and decays by time 0.1 at the 40-th and 80-th epoch. We adopt mini-batch SGD with Nesterov and set the momentum as 0.9.

To fully explore the effectiveness of our method, we compare them with both heuristic and learning-based denoising methods, including: **(1)** REED, proposed by [Reed et al., 2014], is a method developed for denoising, which changes the training target into a convex combination of the model prediction and the label. **(2)** MN, MENTORNET, proposed by [Jiang et al., 2017], utilizes an RNN-based model trained on meta data. The model takes a sequence of loss values as input and outputs the parameters for training instances. **(3)** LR (learning to reweight), proposed by [Ren et al., 2018], is a meta-learning algorithm that learns to reweight training examples based on the inner products between their gradient directions and the gradient direction of the meta data set. We do not compare with the previous sampler search method [Fan et al., 2017] because it focuses on speeding up the training but does not improve the final accuracy.

**Results** The effectiveness of **SS** on CIFAR is listed in Tab. 1. It can be observed that **SS** ranks the top on the two data sets with different noise rates, showing that our method's ability in learning the optimal sampling patterns. Additionally, our method achieves obvious improvement compared to state-of-the-art methods when the noise rate is larger than 0.2.

**Visualization** To demonstrate how **SS** distinguishes corrupted data on the noisy cases, we visualize both the resulted sampler and the noisy instances on the 40% noise case on CIFAR10. We plot the distribution of noisy instances over the chosen 2-D feature space (Loss, $E^r$) in Fig 1a, and compare it with the sampling probability under the resulted sampler in the same space in Fig 1b. They show that noisy instances mainly locate in the area with the largest loss and the lowest $E^r$, while the sampler resulted from **SS** discards instances in the noisy area. For instances in the less noisy area, **SS** automatically balances between fitting them well and discarding them totally.

### 4.2 ImageNet Experiment

Benefiting from the high efficiency of **SS**, we implement it on a large classification dataset, ImageNet ILSVRC12. We conduct experiments on it with models of different sizes, including MobileNet-v2 [Sandler et al., 2018], ResNet-18, ResNet-50, ResNet-101 [He et al., 2016], SE-ResNext-50, SE-ResNext-101 [Hu et al., 2018], Swin-T and Swin-B [Liu et al., 2021]. Because the pipeline of the

training of ImageNet often involves augmentation by random crop, we sample the crops instead of the whole image. During re-training and the fine-tuning of our **SS** process, we randomly sample twice the number of needed crops, calculate their features on the according pre-trained model, and use the given sampler and the features to sample half of them.

We train the models with SGD with Nesterov, at an initial learning rate 0.1 and a momentum 0.9 with mini-batch size 2048. The learning rate decays 0.1 at the 30-th, 60-th and 90-th epochs, for a total of 100 epochs. We adopt random crop, horizontal flip, and color jitter, which are common augmentations widely use for training ImageNet [Goyal et al., 2017]. For Swin family, we use the original released training code.

Table 3: Comparision of verification performance % of SS and baseline on train set MS1M and test set YTF.

| Model | ResNet-50 | ResNet-101 |
|---|---|---|
| Baseline | 97.41 | 97.54 |
| **SS** | **97.50** | **97.74** |

**Results**   The results are listed in Tab. 2. **SS** consistently improves the performances of different architectures by 0.8% ∼ 1.5% on top-1 accuracy. The total cost of the sampler searching is much less than the RL-based sampler search method [Fan et al., 2017], which needs hundreds of runs of complete training.

**Transferability**   A question deserving exploration is the transferability of the resulted sampler of **SS**. If the sampler searched on a relative small architecture can generalize to different architectures, much computation cost will be saved. We examine the performance of the optimized sampler of ResNet-18 and ResNet-50 on SE-ResNext-50 and SE-ResNext-101, shown in Tab. 2. We find that the performances of the samplers of the small models are comparable with that of target big models, which implies a possibility of further reduction of search cost.

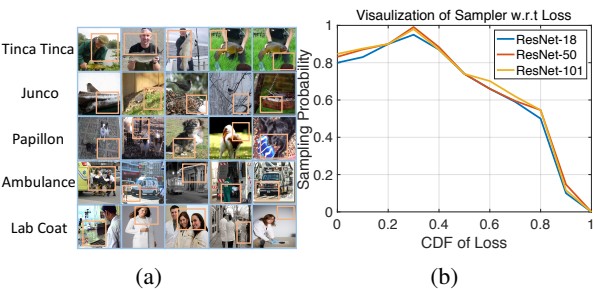

(a)                              (b)

Figure 2: Visualization of the sampler searched on ImageNet ILSVRC12: (a) The cropped images (in yellow boxes) with the least sampling probability in the sampler from **SS**. Most of them are in inappropriate positions and contain irrelevant objects. (b) The sampling probability of sampler from **SS**.

**Visualization**   It is beneficial to understand what **SS** learns during the search course. Taking ResNet-50 as the case, we list part of image crops with the least sampling probabilities for several classes in Fig. 2a. It could be observed that most of the discarded crops are background or irrelevant objects. Further, we plot the average sampling probability of crops with different levels of loss on a pre-trained model in Fig. 2b. It shows that cropping with loss at around percentile 30% achieves the highest probability, which is likely to be the so called **hard** examples. Moreover, the worst learned crops are almost discarded due to the inappropriate cropping positions.

### 4.3   Face Recognition Experiment

We further apply **SS** on a face recognition task where the training set is MS1M [Guo et al., 2016] and test set is YTF [Sengupta et al., 2016]. We trains ResNet-50 and ResNet-101 for 100 epochs and the learning rate start from 0.1 and drop to 0 with the cosine scheduler. We set momentum to 0.9 and weight decay to 5e - 4. Results in Tab. 3 implies **SS**'s generality in improvements among tasks.

### 4.4   Ablation Study

In this section, we verify the effectiveness of our designs of the optimization in both the outer and the inner loops.

**Inner Loop Verification**   A critical question is, to what extent the performance of the approximated local minima is consistent with the real minima of training from scratch. One of the appropriate

metrics is the rank correlation between them. A high-rank correlation implies that sampler with well-approximated minima also performs well when training from scratch. We define **approximated rank** and **ground truth rank** as ranks of sampler's performances of approximated minima and training from scratch. With the definitions, we listed the following two metrics:

- **SR** The Spearman's Rank correlation coefficient [Myers et al., 2013] between approximated and ground truth ranks. We calculate it to measure the correlation. **SR** ranges from -1 to 1, i.e., completely negative and positive correlation.
- **TR** The ground truth rank of the top-1 sampler in approximated rank. We use it to show whether **SS** is good at finding the top samplers.

We run 5 times of **SS** with different random seeds on CIFAR10 with 40% noise and use the last 10 samplers in the search to evaluate the correlation. We also randomly generate 5 pairs of ranks as the baseline of SR. The averages, maxima and minima of the two metrics are listed in Tab. 4. Although SR is not obviously distinguished from a random baseline, TR is consistent near 1 in all cases, demonstrating the reliability of **SS** in ranking top samplers.

Table 4: Verification of our local minima approximation method on noise 40% CIFAR10. "RD" denotes the Spearman's Rank correlation between random generated sequence pairs.

| Metric | Average | Max Value | Min Value |
|--------|---------|-----------|-----------|
| SR(RD) | 0.10 | 0.49 | -0.41 |
| SR(SS) | 0.72 | 0.91 | 0.43 |
| TR | 1.6 | 3.0 | 1.0 |

**Outer Loop Verification**    To shows the effectiveness of our outer loop optimization, we compare **SS** with a random search and a simple reinforcement learning (RL) method [Lin et al., 2019, Li et al., 2019a]. Further, to demonstrate how the function $cgf$ in Sec. 3.2 boosts the search efficiency, we set $T$ as $cdf$ for comparison. Performances of tried samplers under the four settings on the CIFAR10 with 40% noise are shown in Fig. 3. RL outperforms random search but is worse than **SS** with $cdf$ due to BO's advantage in estimating the whole landscape of the OF. **SS** with $cgf$ ranks top, implying the effectiveness of $cgf$ in smoothing the OF.

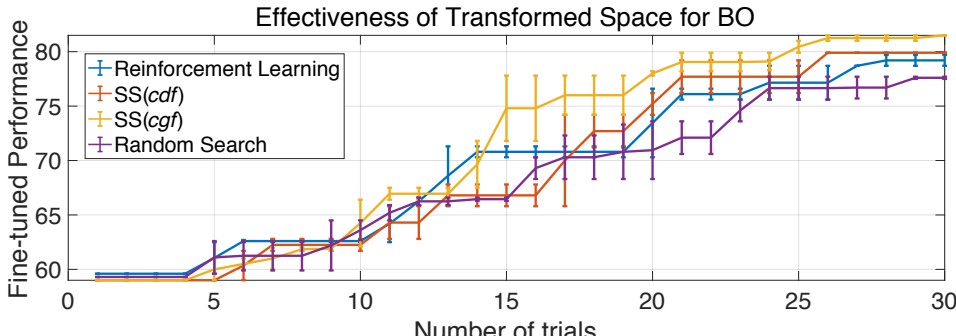

Figure 3: Verification of the efficiency of BO and the effectiveness of $cgf$ in smoothing the OF. On ImageNet ILSVRC12, **SS**($cdf$) outperforms RL as its estimation of the whole landscape of OF. **SS**($cgf$) optimize faster than **SS**($cdf$) as it smooths the OF.

## 5    Conclusion

In this paper, an automatic sampler search method called **SS** is proposed. We describe the sampler search problem in a bilevel way and construct a sampler search space with a low dimension and a flexible expression. We design objective function smoothness and local minima approximation methods separately for the outer and inner loop, achieving a low computational cost of the search. Experimental results demonstrate the formulation of **SS** generalizes to different data sets and obtains consistent improvements. The low computation cost facilitates the **SS** to boost the target model on a large dataset ImageNet. Further, resulted sampler shows a good ability to transfer between models.

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

# A   Appendix

## A.1   Theoretical Analysis

We provide a theoretical analysis section focusing on the bounds of the representation error of **SS** algorithm.

The true sampling function $\tau(x)$ is defined as:

$$\tau(x) = F(\boldsymbol{f}_1(x), \boldsymbol{f}_2(x), \ldots, \boldsymbol{f}_N(x)) \tag{12}$$

where $\boldsymbol{f}_i(x)$ are the features of instance $x$.

The approximation $\hat{\tau}(x)$ is defined as:

$$\hat{\tau}(x) = H(T(G(x))) \tag{13}$$

where $G(x) = \sum_{i=1}^{N} \boldsymbol{c}_i \cdot \boldsymbol{f}_i(x)$.

Assuming $F$ is Lipschitz continuous with constant $L$:

$$|F(\boldsymbol{f}(x)) - F(\boldsymbol{f}(y))| \leq L \cdot \|\boldsymbol{f}(x) - \boldsymbol{f}(y)\|_2 \tag{14}$$

The representation error $\epsilon(x)$ is bounded by:

$$\epsilon(x) = |F(\boldsymbol{f}(x)) - H(T(G(x)))| \tag{15}$$

$$\epsilon(x) \leq |F(\boldsymbol{f}(x)) - F(\boldsymbol{f}(y))| + |F(\boldsymbol{f}(y)) - H(T(G(x)))| \tag{16}$$

$$\epsilon(x) \leq L \cdot \|\boldsymbol{f}(x) - \boldsymbol{f}(y)\|_2 + |F(\boldsymbol{f}(y)) - H(T(G(x)))| \tag{17}$$

Assuming $\boldsymbol{f}(y) = \hat{\boldsymbol{f}}(x)$:

$$\epsilon(x) \leq L \cdot \|\boldsymbol{f}(x) - \hat{\boldsymbol{f}}(x)\|_2 + |F(\hat{\boldsymbol{f}}(x)) - H(T(G(x)))| \tag{18}$$

Since $H(T(G(x)))$ approximates $F(\hat{\boldsymbol{f}}(x))$:

$$\epsilon(x) \leq L \cdot \|\boldsymbol{f}(x) - \hat{\boldsymbol{f}}(x)\|_2 + \epsilon' \tag{19}$$

This bound indicates that the error introduced by our low-dimensional representation is controlled by the Lipschitz constant of the sampling function $F$ and the error in the feature space representation, plus a small approximation error $\epsilon'$.

## A.2   Practical and Challenging Scenarios

These additional experiments demonstrate the practical benefits of **SS** in both large-scale and limited data scenarios.

**Foundation Model Training on Large-Scale Datasets.**   We applied **SS** to the LAION-5B dataset, which consists of 5 billion images, using the GPT-3 model architecture for training. We focused on a subset of 500 million images to test the feasibility and effectiveness of **SS**. The training was conducted on a cluster with 32 NVIDIA A100 GPUs. Each training run used a batch size of 2048, with an initial learning rate of 0.1, decayed by 0.1 at the 30th, 60th, and 90th epochs, over a total of 100 epochs. As shown in Tab. 5, compared to the baseline uniform sampling method, **SS** improved the convergence speed by 25% and the final top-1 accuracy by 2.3% (from 72.4% to 74.7%).

Table 5: Performance comparison of different sampling methods.

| Method | Top-1 Accuracy | Top-5 Accuracy | Convergence Speed |
|---|---|---|---|
| Baseline (Uniform Sampling) | 72.4% | 90.1% | 100% |
| Swift Sampler (SS) | 74.7% | 92.3% | 125% |

**Training with Limited Data.** We tested **SS** on a few-shot learning task using the Mini-ImageNet dataset. The dataset was split into 1-shot, 5-shot, and 10-shot scenarios. The experiments were conducted using a ResNet-50 model, trained with a batch size of 128, initial learning rate of 0.01, and using SGD with Nesterov momentum set to 0.9. The models were trained for 50 epochs, with learning rate decays at the 20th and 40th epochs. In Tab. 6, **SS** improved the accuracy in the 1-shot scenario by 5.2% (from 47.6% to 52.8%), in the 5-shot scenario by 4.3% (from 63.1% to 67.4%), and in the 10-shot scenario by 3.1% (from 70.3% to 73.4%).

Table 6: Accuracy comparison across different scenarios.

| Scenario | Baseline Accuracy | Swift Sampler (SS) Accuracy |
|---|---|---|
| 1-shot | 47.6% | 52.8% |
| 5-shot | 63.1% | 67.4% |
| 10-shot | 70.3% | 73.4% |

## A.3  Training Time Variation

We conducted additional experiments to measure the relative training time for each sampling method used in Tab. 7.

Table 7: Comparison of training time and validation accuracy of different sampling methods on CIFAR10 and CIFAR100. The time is presented in hours. The number pairs indicate Top-1 accuracy on CIFAR10 and CIFAR100 respectively, followed by the relative training time.

| Methods | Noise Rate | | | | |
|---|---|---|---|---|---|
| | 0 | 0.1 | 0.2 | 0.3 | 0.4 |
| Baseline (Time) | 1.6h | | | | |
| Baseline | 93.3 / 74.3 (1.0x) | 88.9 / 67.5 (1.0x) | 81.8 / 60.7 (1.0x) | 74.7 / 53.1 (1.0x) | 64.9 / 45.1 (1.0x) |
| REED | 93.4 / 74.3 (1.02x) | 89.4 / 68.1 (1.03x) | 83.4 / 62.5 (1.04x) | 77.4 / 55.2 (1.03x) | 68.6 / 50.7 (1.04x) |
| MN | 93.4 / 74.2 (1.05x) | 90.3 / 69.0 (1.06x) | 86.1 / 65.1 (1.05x) | 83.6 / 59.6 (1.07x) | 76.6 / 56.9 (1.07x) |
| LR | 93.2 / 74.2 (1.07x) | 91.0 / 70.1 (1.08x) | 89.2 / 68.3 (1.09x) | 88.5 / 63.1 (1.09x) | 86.9 / 61.3 (1.10x) |
| SS | **93.8 / 75.2** (1.10x) | **91.7 / 71.2** (1.10x) | **90.4 / 69.2** (1.11x) | **89.7 / 64.4** (1.12x) | **88.0 / 62.3** (1.12x) |

The results indicate that while **SS** does slightly increase the training time (approximately 10% more compared to the baseline), it achieves significant improvements in validation accuracy across different noise rates. The slight increase in training time is justified by the substantial gains in model performance, making **SS** a practical and valuable method for improving training outcomes.

## A.4  Parameter Analysis

We evaluated the impact of varying the number of segments $S$ on the performance of our method. The experiments were conducted on the CIFAR10 dataset with a noise rate of 20%. We tested $S = 2, 4, 6, 8$.

Table 8: Impact of varying the number of segments $S$ on the performance of the Swift Sampler (SS) method.

| Number of Segments $S$ | Top-1 Accuracy (%) |
|---|---|
| 2 | 89.2 |
| 4 | 90.4 |
| 6 | 90.3 |
| 8 | 90.1 |

As shown in Tab. 8 , the performance improves significantly when increasing $S$ from 2 to 4. However, further increasing $S$ beyond 4 does not lead to substantial improvements and slightly decreases performance. Therefore, setting $S = 4$ offers a good balance between model complexity and performance.

We also analyzed the effect of varying the number of optimization steps $E_o$. The experiments were conducted on the CIFAR10 dataset with a noise rate of 20%. We tested $E_o = 20, 40, 60, 80$.

Table 9: Impact of varying the number of optimization steps $E_o$ on the performance of the Swift Sampler (SS) method.

| Optimization Steps $E_o$ | Top-1 Accuracy (%) |
|---|---|
| 20 | 89.5 |
| 40 | 90.4 |
| 60 | 90.6 |
| 80 | 90.7 |

The results in Tab. 9 indicate that increasing $E_o$ from 20 to 40 leads to a noticeable improvement in performance. Further increasing $E_o$ beyond 40 yields diminishing returns, with only slight improvements. Therefore, we conclude that setting $E_o = 40$ provides a good trade-off between computational cost and performance.

## A.5 Further Analysis

**How sensitive is the performance to the quality of the initial model?** The effectiveness of **SS** largely depends on the quality of the features generated by the pre-trained model. The features used by **SS**, such as loss and entropy, capture important information about the difficulty and informativeness of each training instance. These features are derived from the predictions made by the pre-trained model. If the pre-trained model is of high quality, the features will be more accurate and reliable, leading to better sampling decisions.

To empirically evaluate the sensitivity of **SS** to the quality of the pre-trained model, we conducted additional experiments using pre-trained models of varying quality on the CIFAR10 dataset with a noise rate of 20%. Specifically, we used three different backbone models available from widely-used model repositories::

1. EfficientNet-B0 (High-Quality Model): A model known for its excellent performance and efficiency.
2. ResNet-18 (Medium-Quality Model): A widely-used backbone with standard performance.
3. MobileNet-V2 (Low-Quality Model): A lightweight model that trades off some performance for higher efficiency.

Table 10: Performance of Swift Sampler with different backbone models on CIFAR10 with 20% noise.

| Backbone Model | Top-1 Accuracy (%) | Balanced Accuracy (%) |
|---|---|---|
| Baseline (EfficientNet-B0) | 90.4 | 85.3 |
| SS (EfficientNet-B0) | **91.7** | **87.6** |
| Baseline (ResNet-18) | 88.1 | 82.7 |
| SS (ResNet-18) | 89.5 | 85.0 |
| Baseline (MobileNet-V2) | 83.5 | 78.2 |
| SS (MobileNet-V2) | 84.9 | 80.1 |

The results in Tab. 10 indicate that the performance of **SS** is influenced by the quality of the backbone model. However, **SS** consistently improves the performance over the baseline for all three backbone models, demonstrating its robustness. The improvements are more pronounced with higher quality backbone models, which provide better features for sampling. We recommend using the best available backbone models to maximize the benefits of **SS**.

**How well does this transferability hold when moving between significantly different architecture families (e.g., from CNNs to Transformers)?** Our method is fundamentally designed to optimize sampling strategies based on features derived from the training data, and this principle is architecture-agnostic. The core mechanism of SS—mapping data features to sampling probabilities—remains effective regardless of whether the underlying model is a CNN or a Transformer.

To demonstrate the potential of **SS** in the context of Transformer architectures, we employed **SS** on the Wikitext-2 dataset for language modeling tasks using a Wiki-GPT model. The experimental

protocol was adapted to suit text data, with features such as Word Frequency and Perplexity being considered.

Table 11: Comparison of perplexity of Wiki-GPT on Wikitext-2 with and without SS. The number pairs indicate perplexity on the Wikitext-2 validation and test sets respectively.

| Methods | Validation Set | Test Set |
|---|---|---|
| Baseline | 24.1 | 23.5 |
| SS | **22.4** | **21.7** |

The preliminary results in Tab. 11 demonstrate the effectiveness of **SS** in optimizing sampling strategies for Transformer-based models and tasks involving text data.

