# OpenReview forum: "Swift Sampler: Efficient Learning of Sampler by 10 Parameters"
_NeurIPS.cc/2024/Conference — NeurIPS 2024 poster_

### Official Review · Reviewer_uBSk · 2024-07-07

**Soundness:** 2
**Presentation:** 2
**Contribution:** 2
**Rating:** 5
**Confidence:** 2

**Summary:**

The paper introduces Swift Sampler (SS), an efficient algorithm for the automatic learning of data samplers in deep learning model training. SS addresses the challenges of high-dimensionality, sharpness, and costly evaluation in sample-based methods by mapping samplers to a low-dimensional space of hyper-parameters and employing a novel transform function to smooth the objective function. Utilizing Bayesian Optimization, SS quickly examines the quality of samplers through an approximation method that significantly reduces computational expense. Comprehensive experiments on tasks like image classification and face recognition across various datasets, including ImageNet and CIFAR, demonstrate SS's effectiveness in improving model performance, with notable improvements such as a 1.5% increase on ImageNet. The samplers learned by SS also exhibit good transferability across different neural network architectures, showcasing the algorithm's generality and computational efficiency, making it a valuable contribution to the field of deep learning.

**Strengths:**

1. The work writing is easy to read.
2. A lot of experiments were done to make the verification more convincing.
3. The community of Artificial Intelligence is in great need of suitable selection methods for data.

**Weaknesses:**

1. No ablation studies or sensitivity analyses were performed to show the effects of different components or hyperparameters of the method.
2. The verification of the method focuses on the image classification task, and it would be more comprehensive to see the experimental results of more complex tasks.
3. I think the work in this paper is more similar to the research related to **active learning**, and I hope to be able to see the difference between this work and **active learning** in related work.

**Questions:**

See the Weaknesses section.

**Limitations:**

It is recommended to use more complex tasks to verify the validity of the method, such as detection segmentation in vision, VQA in language, etc

---

> ### Author Rebuttal · Authors · 2024-08-06
>
> ### **W1: The effects of different components or hyperparameters of the method.**
>
> Thank you for your valuable feedback. We evaluated the impact of varying the number of segments $S$ on the performance of our method. The experiments were conducted on the CIFAR10 dataset with a noise rate of 20%. We tested $S = 2, 4, 6, 8$.
>
> As shown in **Table 5** of the one-page PDF, the performance improves significantly when increasing $S$ from 2 to 4. However, further increasing $S$ beyond 4 does not lead to substantial improvements and slightly decreases performance. Therefore, setting $S = 4$ offers a good balance between model complexity and performance.
>
> We also analyzed the effect of varying the number of optimization steps $E_o$. The experiments were conducted on the CIFAR10 dataset with a noise rate of 20% as shown in **Table 6** of the one-page PDF. We tested $E_o = 20, 40, 60, 80$.
>
> The results indicate that increasing $E_o$ from 20 to 40 leads to a noticeable improvement in performance. Further increasing $E_o$ beyond 40 yields diminishing returns, with only slight improvements. Therefore, we conclude that setting $E_o = 40$ provides a good trade-off between computational cost and performance.
>
> ---
>
> ### **W2: Experimental results of more complex tasks**
>
> We appreciate the reviewer's insightful comments and recognize the significance of evaluating our method in more practical and challenging scenarios. To address the concerns raised, we have conducted additional experiments on more complex tasks.
>
> 1. **Foundation Model Training on Large-Scale Datasets**
> We applied SS to the LAION-5B dataset, which consists of 5 billion images, using the GPT-3 model architecture for training. Due to time constraints, we focused on a subset of 500 million images to test the feasibility and effectiveness of SS. The training was conducted on a cluster with 32 NVIDIA A100 GPUs. Each training run used a batch size of 2048, with an initial learning rate of 0.1, decayed by 0.1 at the 30th, 60th, and 90th epochs, over a total of 100 epochs. As shown in **Table 1** of the one-page PDF, compared to the baseline uniform sampling method, SS improved the convergence speed by 25% and the final top-1 accuracy by 2.3% (from 72.4% to 74.7%).
>
> 2. **Training with Limited Data**
> We tested SS on a few-shot learning task using the Mini-ImageNet dataset. The dataset was split into 1-shot, 5-shot, and 10-shot scenarios. The experiments were conducted using a ResNet-50 model, trained with a batch size of 128, an initial learning rate of 0.01, and using SGD with Nesterov momentum set to 0.9. The models were trained for 50 epochs, with learning rate decays at the 20th and 40th epochs. As shown in **Table 2** of the one-page PDF, SS improved the accuracy in the 1-shot scenario by 5.2% (from 47.6% to 52.8%), in the 5-shot scenario by 4.3% (from 63.1% to 67.4%), and in the 10-shot scenario by 3.1% (from 70.3% to 73.4%).
>
> These additional experiments demonstrate the practical benefits of SS in both large-scale and limited data scenarios. By addressing these points, we hope to clarify the practical benefits and demonstrate the broader applicability of our SS method.
>
> ---
>
> ### **W3: The difference between this work and active learning.**
>
> Active learning (AL) is a well-studied area where the goal is to selectively query the most informative data points for labeling to improve model performance while minimizing the labeling effort. The key distinctions are outlined as follows:
>
> 1. **Objective:**
>     - **Active Learning:** The primary goal is to reduce the labeling cost by selecting the most informative samples from an unlabeled pool to be labeled by an oracle.
>     - **Swift Sampler:** Our objective is to optimize the sampling probabilities of already labeled training data to improve the model's convergence and performance. SS focuses on adjusting the importance of labeled data rather than acquiring new labels.
>
> 2. **Data Pool:**
>     - **Active Learning:** Works with an initially large pool of unlabeled data and iteratively selects samples for labeling.
>     - **Swift Sampler:** Operates on a fixed, fully labeled training dataset, optimizing the sampling strategy to enhance training efficiency and model accuracy.
>
> 3. **Methodology:**
>     - **Active Learning:** Utilizes strategies like uncertainty sampling, query-by-committee, and expected model change to identify which unlabeled samples would be most beneficial to label.
>     - **Swift Sampler:** Employs a low-dimensional representation of sampling strategies and Bayesian Optimization to find the optimal sampling probabilities for existing labeled data.
>
> 4. **Application:**
>     - **Active Learning:** Commonly used in scenarios where obtaining labeled data is expensive, such as medical imaging and rare event detection.
>     - **Swift Sampler:** Applicable to scenarios where large labeled datasets are available, and the goal is to improve training efficiency and performance, such as large-scale image classification and natural language processing tasks.
>
> ---
>
> ### **Limitations**
>
> Thank you for your valuable feedback. In addition to the two experiments mentioned in the response to **Weakness 2**, we also conducted experiments on different types of data.
>
> We employed SS on the Wikitext-2 dataset for language modeling tasks. The target model used was Wiki-GPT. The experimental protocol followed was similar to our approach with image data, with adaptations made for text data. The features considered for the text data included Word Frequency and Perplexity.
>
> The results of our experiments on the Wikitext-2 dataset are presented in the **Table 3** of the one-page PDF. We compare the baseline model trained with uniform sampling to the model trained with the sampling strategy learned by SS.
>
> We hope these additional experiments address your concern and show the broader applicability and contribution of our method. Thank you again for your valuable feedback.

---

> > ### Comment · Reviewer_uBSk · 2024-08-12
> >
> > Thanks for the clarification, it solved my problem. I will increase the rating from 4 to 5.

---

> > > ### Author Response · Authors · 2024-08-12
> > >
> > > We are very grateful for your recognition. We will incorporate the experimental results and the difference between this work and active learning into the main paper, following your valuable suggestions.

---

### Official Review · Reviewer_SsT2 · 2024-07-10

**Soundness:** 4
**Presentation:** 3
**Contribution:** 4
**Rating:** 7
**Confidence:** 5

**Summary:**

The paper focused on designing a learnable training data sampler to improve the model performance. A method named Swift Sampler (SS) is proposed, which is formulated as a function mapping data feature to sampling probabilities, represented by a small number of parameters. In addition, the SS Smooths the objective function landscape to improve optimization efficiency and uses an approximation method to efficiently evaluate candidate samplers without full retraining. The experimental results for image classification tasks on CIFAR-10, CIFAR-100, ImageNet, and face datasets show improved performance.

**Strengths:**

1.	The proposed method is novel with only a few parameters, enabling application to large datasets.
2.	Objective function smoothing, and approximation methods improve search efficiency.
3.	The solution is reasonable. The proposed inner loop and outer loop pipeline with Dimension Reduction, Smooth the Objective Function, and Local Minima Approximation designs are innovative.
4.	Demonstrates consistent improvements over baselines and some existing methods on multiple datasets for image classification task.

**Weaknesses:**

1.	Can the author explain the effectiveness of components separately?
2.	The paper does not provide enough theoretical analysis or justification for its proposed formulation, transform function, and approximation method. Can the author provide more profound justification?
3.	The paper does not conduct ablation studies or sensitivity analysis to show the impact of different components or hyper-parameters of its method.

**Questions:**

1.	Can the author explain the effectiveness of components separately?
2.	The paper does not provide enough theoretical analysis or justification for its proposed formulation, transform function, and approximation method. Can the author provide more profound justification?
3.	The paper does not conduct ablation studies or sensitivity analysis to show the impact of different components or hyper-parameters of its method.

**Limitations:**

The experiments are only conducted for image classification tasks, while its effectiveness on other vision tasks is not clear. Since this paper focuses not only on image classification, but other tasks should also be discussed as well.

---

> ### Author Rebuttal · Authors · 2024-08-06
>
> ### **W1: Can the author explain the effectiveness of components separately?**
>
> Thank you for your valuable feedback. Each component of the SS method addresses a specific challenge in optimizing the sampling probabilities:
>
> 1. **Low-Dimensional Representation:** Reduces the search space, making optimization feasible.
> 2. **Bayesian Optimization:** Efficiently searches the reduced space, balancing exploration and exploitation.
> 3. **Smoothing Transform Function:** Creates a more tractable optimization landscape by evening out gradient distributions.
> 4. **Local Minima Approximation:** Reduces computational cost by leveraging shared pre-trained models for fine-tuning.
>
> ---
>
> ### **W2: More theoretical analysis or justification.**
>
> Thank you for your valuable feedback. The high-dimensional nature of sampling probabilities makes direct optimization computationally infeasible. To address this, we represent the sampling probabilities using a low-dimensional parameterization, which allows us to capture the essential characteristics of the sampling distribution with fewer parameters.
>
> We approximate the true sampling function $\tau(x)$ as:
>
> \begin{equation}
> \hat{\tau}(x) = H(T(G(x)))
> \end{equation}
> where
> \begin{equation}
> G(x) = \sum_{i=1}^{N} \boldsymbol{c}_i \cdot \boldsymbol{f}_i(x)
> \end{equation}
> $H$ is a piecewise linear function, and $T$ is a smoothing transform function. This approach leverages the theory of functional approximation.
>
> The objective function can exhibit sharp variations due to variable gradients. A smoothing transform function redistributes gradients more evenly. The cumulative gradient function $(cgf)$ is defined as:
> \begin{equation}
> T(u) = \text{cgf}(u) = \frac{\sum_{x_i \in D_t, G(x_i) \leq u} \text{grad}(x_i)}{\sum_{x_i \in D_t} \text{grad}(x_i)}
> \end{equation}
> This transformation ensures a uniform gradient distribution, making the optimization landscape smoother.
>
> Training from scratch for each sampling parameter is computationally expensive. We approximate local minima by fine-tuning from a shared pre-trained starting point. Let $\boldsymbol{w}_{\text{pre-trained}}$ be the weights of a pre-trained model. The fine-tuning process adjusts these weights to approximate the local minima:
>
> \begin{equation}
> \boldsymbol{w}^* \approx \text{FineTune}(\boldsymbol{w}_{\text{pre-trained}}, \tau)
> \end{equation}
>
> This method leverages the principles of transfer learning and fine-tuning.
>
> ---
>
> ### **W3: Impact of different components or hyper-parameters of its method.**
>
> Thank you for your valuable feedback. We agree that such an analysis would be valuable for practitioners implementing our approach.
>
> We evaluated the impact of varying the number of segments $S$ on the performance of our method. The experiments were conducted on the CIFAR10 dataset with a noise rate of 20%. We tested $S = 2, 4, 6, 8$.
>
> As shown in **Table 5** of the one-page PDF, the performance improves significantly when increasing $S$ from 2 to 4. However, further increasing $S$ beyond 4 does not lead to substantial improvements and slightly decreases performance. Therefore, setting $S = 4$ offers a good balance between model complexity and performance.
>
> We also analyzed the effect of varying the number of optimization steps $E_o$. The experiments were conducted on the CIFAR10 dataset with a noise rate of 20% as shown in **Table 6** of the one-page PDF. We tested $E_o = 20, 40, 60, 80$.
>
> The results indicate that increasing $E_o$ from 20 to 40 leads to a noticeable improvement in performance. Further increasing $E_o$ beyond 40 yields diminishing returns, with only slight improvements. Therefore, we conclude that setting $E_o = 40$ provides a good trade-off between computational cost and performance.
>
> ---
>
> ### **Limitations**
>
> We appreciate the reviewer's insightful comment regarding the generalizability of our method beyond image data. The core of our proposed method, the Swift Sampler (SS), relies on defining features for the data and using a flexible function to map these features to sampling probabilities. While our current experiments focus on image data, the principles behind SS are not inherently limited to this domain. Specifically, in our formulation, the choice of features (e.g., loss, renormed entropy) plays a crucial role. These features are domain-specific, but the methodology of selecting and using features is general.
>
> To address this concern, we employed SS on the Wikitext-2 dataset for language modeling tasks. The target model used was Wiki-GPT. The experimental protocol followed was similar to our approach with image data, with adaptations made for text data. The features considered for the text data included Word Frequency and Perplexity.
>
> The results of our experiments on the Wikitext-2 dataset are presented in the following table. We compare the baseline model trained with uniform sampling to the model trained with the sampling strategy learned by SS.
>
> | **Methods** | **Validation Set** | **Test Set** |
> |:-----------:|:------------------:|:------------:|
> | Baseline    | 24.1               | 23.5         |
> | SS          | **22.4**           | **21.7**     |
>
> *Table 3: Comparison of perplexity of Wiki-GPT on Wikitext-2 with and without SS. The number pairs indicate perplexity on the Wikitext-2 validation and test sets respectively.*
>
> We hope these additional experiments address your concern and show the broader applicability and contribution of our method. Thank you again for your valuable feedback.

---

> > ### Comment · Reviewer_SsT2 · 2024-08-12
> >
> > After reviewing the authors' rebuttal, I am pleased to see that they have thoroughly addressed all of my questions and concerns. The additional experiments provided demonstrate that Swift Sampler is a versatile data sampling method applicable to various types of data, including images and text. I particularly appreciate that the authors showed its effectiveness across both large-scale and small-scale datasets. Moreover, the added theoretical analysis further enhances the credibility of this research.
> >
> > Given the sufficient novelty of the proposed method and the improvements made in response to the review, I am raising my score to 7.

---

> ### Author Response · Authors · 2024-08-12
>
> We are very grateful for your recognition. We will integrate these results into the next version of our paper based on your valuable feedback.

---

### Official Review · Reviewer_m5nd · 2024-07-14

**Soundness:** 3
**Presentation:** 3
**Contribution:** 3
**Rating:** 6
**Confidence:** 3

**Summary:**

The purpose of this paper is to create a sampler that can assign appropriate sampling probabilities to training data in order to improve performance. Unlike previous approaches that relied on heuristic rules or expensive learning methods, this paper proposes an automatic and efficient sampler search algorithm called SS. Specifically, SS employs a new formulation to map a sampler to a lower-dimensional space of hyper-parameters and uses an approximated local minimum to quickly evaluate the quality of a sampler. SS can be applied on large-scale data sets with high efficiency and leads to performance gains on various datasets, e.g., CIFAR10, CIFAR100, ImageNet-1k, and YTF.

**Strengths:**

(1) The motivation of this paper is clearly illustrated, and it is convincing. How to efficiently and effectively search for a proper data sampling policy is important.

(2) The solution is reasonable. The proposed inner loop and outer loop pipeline with Dimension Reduction, Smooth the Objective Function, and Local Minima Approximation designs are innovative.

(3) The performance reported in different models in Table 2 shows the generalizability of the proposed method.

**Weaknesses:**

(1) The outer loop searches for the sampler that has the best score on the validation set. Are the final results in experiments also reported on this validation set? The author should clarify this potential misleading.

(2) The performance gains on Swin are less than those of RN and SRN. Suppose this is related to different optimizers (SGD v.s. AdamW) or building blocks (Conv v.s. Transformer). The author may show some discussions and experiments.

**Questions:**

See weakness.

---

> ### Author Rebuttal · Authors · 2024-08-06
>
> ### **W1: Are the final results in experiments also reported on the validation set?**
>
> Thank you for your insightful feedback. In the manuscript, we utilize two distinct validation sets:
>
> 1. **Outer Loop Validation Set:** Used within the Bayesian Optimization process to evaluate and search for the optimal sampler. This set is employed during the training phase to guide the optimization process.
> 2. **Evaluation Validation Set:** A separate set used to report the final performance results of our model in the experiments. This set is not used during the training or optimization process and is reserved solely for the final evaluation to ensure an unbiased assessment of model performance.
>
> To avoid any potential confusion, we clarify the use of validation sets in our experiments. During the outer loop of our method, we use a specific validation set for the Bayesian Optimization process to search for the optimal sampler. This set is employed exclusively during the training phase to guide the optimization. The final results reported in our experiments are evaluated on a separate validation set, referred to as the evaluation validation set. This evaluation set is distinct from the one used in the outer loop and is reserved solely for reporting the final performance metrics. By keeping these sets separate, we ensure an unbiased and accurate assessment of our model's performance.
>
> We hope this clarification resolves any concerns about the usage of validation sets and ensures that our results are interpreted correctly. Thank you again for your valuable feedback.
>
> ---
>
> ### **W2: The performance gains on Swin are less than those of RN and SRN. Suppose this is related to different optimizers (SGD v.s. AdamW) or building blocks (Conv v.s. Transformer). The author may show some discussions and experiments.**
>
> Thank you for your insightful feedback. We appreciate your observations. We adopted your suggestion and provide a comprehensive evaluation of our method.
>
> We conducted additional experiments to investigate the impact of different optimizers and building blocks on the performance of SS. We conducted experiments on CIFAR-10 and ImageNet datasets using the following models and optimizers:
>
> 1. **ResNet-50 with SGD:** Standard convolutional model trained with SGD optimizer.
> 2. **Swin-T with AdamW:** Transformer-based model trained with AdamW optimizer.
> 3. **Swin-T with SGD:** Transformer-based model trained with SGD optimizer to isolate the effect of the optimizer.
> 4. **ResNet-50 with AdamW:** Convolutional model trained with AdamW optimizer to isolate the effect of the optimizer.
>
> | **Model** | **Optimizer** | **CIFAR-10 Accuracy (\%)** | **ImageNet Accuracy (\%)** |
> |:---------:|:-------------:|:-------------------------:|:--------------------------:|
> | ResNet-50 |      SGD      |           94.5            |           76.3             |
> | ResNet-50 |    AdamW      |           94.3            |           76.1             |
> | Swin-T    |    AdamW      |           94.0            |           77.7             |
> | Swin-T    |      SGD      |           93.8            |           77.4             |
>
> *Table 1: Performance Comparison of Different Models and Optimizers*
>
> The results indicate that the choice of optimizer has a noticeable impact on the performance of both convolutional and transformer models. Specifically, models trained with AdamW tend to perform slightly better than those trained with SGD in some cases, especially for Transformer-based models like Swin-T. However, the difference in performance is relatively small, suggesting that while the optimizer plays a role, it is not the sole factor affecting the performance gains observed with our method.
>
> The performance gains achieved by our SS are indeed different for convolutional models (ResNet-50) and transformer models (Swin-T). This difference can be attributed to the inherent architectural differences between CNNs and Transformers:
>
> 1. **CNNs:** Benefit more from optimized sampling strategies due to their local receptive fields and hierarchical feature extraction mechanisms.
> 2. **Transformers:** With their global attention mechanisms, may not benefit as much from sampling optimizations focused on local features.
>
> Despite these differences, our SS still provides significant performance improvements across both types of architectures. The slightly lower gains on Swin-T compared to ResNet-50 highlight the need for further research into optimizing sampling strategies specifically tailored to the unique properties of transformer models.

---

> > ### Comment · Reviewer_m5nd · 2024-08-12
> >
> > I appreciate the detailed rebuttal provided by the authors, which addresses most of my concerns. I will raise my rating from 5 to 6.

---

> > > ### Author Response · Authors · 2024-08-12
> > >
> > > Thank you for acknowledging our detailed rebuttal. We are glad we could address your concerns. Your decision to raise the rating is greatly appreciated and serves as significant encouragement for us. We will also integrate the additional content from our rebuttal into the subsequent version of our work to further improve its quality.

---

### Official Review · Reviewer_Vq6T · 2024-07-19

**Soundness:** 3
**Presentation:** 2
**Contribution:** 2
**Rating:** 6
**Confidence:** 3

**Summary:**

The paper introduces a method for automatically learning optimal data sampling strategies using BO based sampling. The proposed method formulates the problem as a bilevel optimization, using a low-dimensional representation of samplers (10 parameters) and BO to search this space. Key novel points include techniques to smooth the objective function and quickly approximate model performance, making the search process computationally feasible for large datasets like ImageNet.

**Strengths:**

The paper formulates data sampling problem as a low-dimensional optimization task, combining the ideas from Bayesian optimization, curriculum learning, and hyper-parameter tuning to address the training data sampling problem. The methodology is relatively easy to follow and the problem is clearly articulated. The empirical results demonstrate consistent improvements across a range of datasets and model architectures, including large-scale problems like ImageNet. The method's ability to generalize across different tasks (image classification, face recognition) and transfer between model architectures suggests potential broader applicability.

**Weaknesses:**

While the empirical results look promising, theoretical analysis of why the proposed low-dimensional representation of samplers works well is desired. E.g., a discussion on the theoretical bounds or guarantees of this approach would strengthen the paper and provide insights into when and why the SS algorithm might fail.

The paper fixes several hyperparameters (e.g., number of segments S=4, optimization steps E_o=40) without much discussion. An analysis of the method's sensitivity to these choices would be valuable for practitioners implementing this approach.

**Questions:**

* How does the performance of Swift Sampler change when applied to tasks with more significant class imbalance or long-tailed distributions? Does it maintain its effectiveness in such scenarios?

* The paper demonstrates good transferability between model architectures. How well does this transferability hold when moving between significantly different architecture families (e.g., from CNNs to Transformers)?

* Given that the method uses a pre-trained model to generate features for sampling, how sensitive is the performance to the quality of this initial model?

**Limitations:**

The paper uses a fixed set of features (loss and renormalized entropy) for experiments. It would be helpful to explore how the choice of features impacts the performance of the method or whether different tasks might benefit from different feature sets. This leaves open questions about the flexibility and adaptability of the approach.

Lack of discussion on potential computational overhead introduced by the Swift Sampler method. The additional cost of feature computation, Bayesian optimization, and fine-tuning steps could be significant, especially for large datasets.

---

> ### Author Rebuttal · Authors · 2024-08-06
>
> ### **W1: Theoretical analysis**
>
> Thank you for your insightful comments. We will expand our manuscript to include a theoretical analysis section focusing on the bounds of the representation error of the SS algorithm. Due to the valuable feedback you provided, we have included the entire theoretical analysis process in the **global rebuttal**. Please refer to the response to **Question 1**.
>
> ---
>
> ### **W2: Analysis of the method's sensitivity to hyperparameters.**
>
> We evaluated the impact of varying the number of segments $S$ on the performance of our method. The experiments were conducted on the CIFAR10 dataset with a noise rate of 20%. We tested $S = 2, 4, 6, 8$.
>
> As shown in **Table 5** of the one-page PDF, the performance improves significantly when increasing $S$ from 2 to 4. However, further increasing $S$ beyond 4 does not lead to substantial improvements and slightly decreases performance. Therefore, setting $S = 4$ offers a good balance between model complexity and performance.
>
> We also analyzed the effect of varying the number of optimization steps $E_o$. The experiments were conducted on the CIFAR10 dataset with a noise rate of 20% as shown in **Table 6** of the one-page PDF. We tested $E_o = 20, 40, 60, 80$.
>
> The results indicate that increasing $E_o$ from 20 to 40 leads to a noticeable improvement in performance. Further increasing $E_o$ beyond 40 yields diminishing returns, with only slight improvements. Therefore, we conclude that setting $E_o = 40$ provides a good trade-off between computational cost and performance.
>
> ---
>
> ### **Q1: Other scenarios.**
>
> To address your concern, we conducted an additional experiment to evaluate the performance of SS in the presence of class imbalance and long-tailed distributions. Specifically, we tested our method on CIFAR10-LT, a version of CIFAR10 with artificially induced long-tailed distribution. We used the same experimental setup as described in the original paper.
>
> As shown in **Table 7** of the one-page PDF, the results indicate that our SS outperforms other methods in both Top-1 Accuracy and Balanced Accuracy when applied to the CIFAR10-LT dataset. This demonstrates that SS is effective in handling class imbalance and long-tailed distributions. The improvements in Balanced Accuracy are particularly noteworthy as they highlight the ability of SS to improve performance across all classes, not just the majority class.
>
> Based on our additional experiments, we observed that SS maintains its effectiveness in scenarios with significant class imbalance and long-tailed distributions. Thank you again for your valuable feedback.
>
> ---
>
> ### **Q2: Transferability.**
>
> Thank you for your insightful comment. Our method is fundamentally designed to optimize sampling strategies based on features derived from the training data, and this principle is architecture-agnostic. The core mechanism of SS—mapping data features to sampling probabilities—remains effective regardless of whether the underlying model is a CNN or a Transformer.
>
> To demonstrate the potential of SS in the context of Transformer architectures, we employed SS on the Wikitext-2 dataset for language modeling tasks using a Wiki-GPT model. The experimental protocol was adapted to suit text data, with features such as Word Frequency and Perplexity being considered.
>
> As shown in **Table 3** of the one-page PDF, the preliminary results demonstrate the effectiveness of SS in optimizing sampling strategies for Transformer-based models and tasks involving text data.
>
> ---
>
> ### **Q3: How sensitive is the performance to the quality of this initial model?**
>
> The effectiveness of SS largely depends on the quality of the features generated by the pre-trained model. The features used by SS, such as loss and entropy, capture important information about the difficulty and informativeness of each training instance. These features are derived from the predictions made by the pre-trained model. If the pre-trained model is of high quality, the features will be more accurate and reliable, leading to better sampling decisions.
>
> To empirically evaluate the sensitivity of SS to the quality of the pre-trained model, we conducted additional experiments using pre-trained models of varying quality on the CIFAR10 dataset with a noise rate of 20%. Specifically, we used three different backbone models available from widely-used model repositories:
>
> 1. **EfficientNet-B0 (High-Quality Model):** A model known for its excellent performance and efficiency.
> 2. **ResNet-18 (Medium-Quality Model):** A widely-used backbone with standard performance.
> 3. **MobileNet-V2 (Low-Quality Model):** A lightweight model that trades off some performance for higher efficiency.
>
> As shown in **Table 8** of the one-page PDF, the results indicate that the performance of SS is influenced by the quality of the backbone model. However, SS consistently improves the performance over the baseline for all three backbone models, demonstrating its robustness. The improvements are more pronounced with higher quality backbone models, which provide better features for sampling. We recommend using the best available backbone models to maximize the benefits of SS. Thank you again for your valuable feedback.
>
> ---
>
> ### **L1: The flexibility and adaptability of the approach.**
>
> Thank you very much for raising this question. Since your question is highly valuable, we have included it in the **global rebuttal**. Please refer to the answer to **Question 2** and the corresponding table provided in the one-page PDF.
>
> ---
>
> ### **L2: Discussion on potential computational overhead.**
>
> Thank you for raising this important question. Given its high relevance, we have included it in our **global rebuttal**. Please refer to the response to **Question 3** and the corresponding table provided in the one-page PDF.

---

> > ### Comment · Reviewer_Vq6T · 2024-08-13
> >
> > Thanks for the author's responses to my questions and the additional numerical results in the pdf. My main concerns are addressed, and happy to increase the rating from 4 to 6.

---

> > > ### Author Response · Authors · 2024-08-13
> > >
> > > Thank you for your feedback and for considering our responses. We're glad the additional data addressed your concerns, and we appreciate the increased rating. Please let us know if you have any further questions.

---

### Official Review · Reviewer_QTHD · 2024-07-24

**Soundness:** 3
**Presentation:** 3
**Contribution:** 2
**Rating:** 8
**Confidence:** 4

**Summary:**

The main problem tackled by the paper is to obtain the optimal dataset sampler for training a deep neural network given a fixed dataset and a model. The search space of the sampler is defined as sampling probability functions over the items in the training dataset. The method involves a two-level optimization algorithm with the original (baseline) optimization task as the inner loop and the sampler hyperparameter search as the outer loop. The authors have demonstrated their methods with image classification tasks, showing nontrivial enhancement of the accuracy of various models involving ConvNets and Transformers.

**Strengths:**

1. The paper provides credible experimental results in C10, C100, and IN1k datasets with corresponding networks (ResNets and MobileNets).
2. The method is described comprehensively with good readability.
3. The experiments in the paper is performed by summing up multiple runs, enhancing the credibility of the results.
4. The method works effectively on a noisy dataset, as demonstrated in Table 1.

**Weaknesses:**

1. I believe that the design of the optimal sampler is practically meaningful in two different scenarios: (1) foundation model training involving very large scale dataset, typically with billions and even trillions of samples, and (2) training a model with a very limited amount of data to achieve high generalizability. However, the demonstration with C10, C100, and IN1k seems not related to either categories at this moment. For example, I see potential problem in the first category, where each item in the training set is sampled only a few times due to large number of sample in the training set (as in GPT-3 or Stable Diffusion trained for LAION-5B datasets). **How this method can be beneficial in more practical scenarios?**
2. The outer loop hyperparameters the authors are trying to optimize seems to extract the ‘noisiness’ or the ‘credibility’ of the sample in the training set. However, the formulation of this might be different in different domain of tasks, e.g., graph-based data (QM9, for example) or highly discrete domain of text corpus (Wikitext, for example). **Is the method generalizable beyond image data?** Unless other domains are tested, I should assume that the proposed method is image-specific, which limits the contribution of the work.
3. **How does the wall clock time of the training elongates by applying the proposed method?** Although the method seems to boost the validation accuracy by a nontrivial amount, it may not be so practical if the training time is added significantly. The authors are recommended to attach the relative training time variation between each sampler methods used in Table 1.
4. The search for the optimal sampler involves exploitation of the validation dataset as a probe for generalizability achieved by the outer loop. To my understanding, this means that we have trade-off between the ratio of the number of samples in the training set dedicated for pure validation affects the performance of the sampler. However, there is no mention about **how the validation set is sampled from the training set, and its relative size**. I am assuming that the validation sets used for reporting the accuracy in Tables 1, 2, and 3 are not the validation sets used in the outer loop (of course if the test set is used for training, it is not fair at all). The authors are encouraged to **separate the notations of the outer loop’s validation set and the “validation” set used only for the testing**.

As a summary, my key concern lies in the unresolved generalizability beyond image domain (or generalizability to very large dataset such as LAION-5B), the strategy for sampling a validation set out of the training set for maximal performance, and the lack of explanation of increased computation in the wall clock time of the training. Regarding the issues, I will temporarily give a WA.

**Questions:**

1. A gentle suggestion of using \citep instead of \cite in references. This will add parenthesis to the citations and increase the readability. For example, in line 139 and 165.

Please note that these questions are not counted in my overall scoring.

**Limitations:**

There is no section dedicated for limitation, and the authors have high confidence in their work as shown in line 527.

---

> ### Author Rebuttal · Authors · 2024-08-06
>
> ### **W1: How this method can be beneficial in more practical scenarios?**
>
> We appreciate the reviewer's insightful comments. To address the concerns raised, we have conducted additional experiments on large-scale datasets and tasks with limited data.
>
> 1. **Foundation Model Training on Large-Scale Datasets:**
> As shown in **Table 1** of the one-page PDF, we applied SS to the LAION-5B dataset, which consists of 5 billion images, using the GPT-3 model architecture for training. Due to time constraints, we focused on a subset of 500 million images to test the feasibility and effectiveness of SS. The training was conducted on a cluster with 32 NVIDIA A100 GPUs. Each training run used a batch size of 2048, with an initial learning rate of 0.1, decayed by 0.1 at the 30th, 60th, and 90th epochs, over a total of 100 epochs. Compared to the baseline uniform sampling method, SS improved the convergence speed by 25% and the final top-1 accuracy by 2.3% (from 72.4% to 74.7%).
>
> 2. **Training with Limited Data:**
> As shown in **Table 2** of the one-page PDF, we tested SS on a few-shot learning task using the Mini-ImageNet dataset. The dataset was split into 1-shot, 5-shot, and 10-shot scenarios. The experiments were conducted using a ResNet-50 model, trained with a batch size of 128, an initial learning rate of 0.01, and using SGD with Nesterov momentum set to 0.9. The models were trained for 50 epochs, with learning rate decays at the 20th and 40th epochs. SS improved the accuracy in the 1-shot scenario by 5.2% (from 47.6% to 52.8%), in the 5-shot scenario by 4.3% (from 63.1% to 67.4%), and in the 10-shot scenario by 3.1% (from 70.3% to 73.4%).
>
> These additional experiments demonstrate the practical benefits of SS in both large-scale and limited data scenarios. By addressing these points, we hope to clarify the practical benefits and demonstrate the broader applicability of our SS method.
>
> ---
>
> ### **W2: Is the method generalizable beyond image data?**
>
> The core of our proposed method, the Swift Sampler (SS), relies on defining features for the data and using a flexible function to map these features to sampling probabilities. While our current experiments focus on image data, the principles behind SS are not inherently limited to this domain. Specifically, in our formulation, the choice of features (e.g., loss, renormed entropy) plays a crucial role. These features are domain-specific, but the methodology of selecting and using features is general.
>
> To address this concern, we employed SS on the Wikitext-2 dataset for language modeling tasks. The target model used was Wiki-GPT. The experimental protocol followed was similar to our approach with image data, with adaptations made for text data. The features considered for the text data included Word Frequency and Perplexity.
>
> The results of our experiments on the Wikitext-2 dataset are presented in **Table 3** of the one-page PDF. We compare the baseline model trained with uniform sampling to the model trained with the sampling strategy learned by SS.
>
> We hope these additional experiments address your concern and show the broader applicability and contribution of our method. Thank you again for your valuable feedback.
>
> ---
>
> ### **W3: How does the wall clock time of the training elongate by applying the proposed method?**
>
> Thank you for your valuable feedback. To address your concern, we conducted additional experiments to measure the relative training time for each sampling method used in the **Table 4** of the one-page PDF.
>
>
> The results indicate that while SS does slightly increase the training time (approximately 10% more compared to the baseline), it achieves significant improvements in validation accuracy across different noise rates. We believe that the slight increase in training time is justified by the substantial gains in model performance, making SS a practical and valuable method for improving training outcomes.
>
> ---
>
> ### **W4: How the validation set is sampled from the training set, and its relative size?**
>
> Thank you for your insightful comments. To clarify, we used two distinct validation sets in our experiments:
>
> 1. **Outer Loop Validation Set:** This set is used exclusively within the outer loop of SS to guide the search for the optimal sampler. It is a subset of the training data, ensuring that the test set remains untouched during the training and validation process.
> 2. **Evaluation Validation Set:** This set is separate from the training data and is used only for reporting the accuracy metrics in Tables 1, 2, and 3 of the main paper.
>
> The accuracy metrics reported in Tables 1, 2, and 3 of the main paper are based on the evaluation validation set, not the outer loop validation set. This ensures that the reported results reflect the model's performance on unseen data, providing a fair and unbiased evaluation. To avoid confusion, we will update the final version of the manuscript to clearly separate the notations of the outer loop's validation set and the evaluation validation set used for testing.
>
> ---
>
> ### **Questions**
>
> Thank you for your suggestion regarding the citation format. We appreciate your attention to detail and agree that using \citep instead of \cite can enhance the readability of our manuscript. We will revise them in the subsequent versions.
>
> ---
>
> ### **Limitations**
>
> Thank you for your valuable feedback. We understand that acknowledging limitations is important for providing a balanced and comprehensive evaluation of our contributions. Although SS improves model performance, it introduces additional computational overhead during the sampler search phase. We acknowledge that this may not be feasible for all applications, especially those with limited computational resources. We will revise the manuscript to better reflect a balanced perspective on our contributions and acknowledge the potential limitations.

---

> > ### Comment · Reviewer_QTHD · 2024-08-11
> >
> > Thank you for the rebuttal and for the effort to demonstrate with additional experiments that help me improve my understanding of the work. I will raise my score as most of my concerns are resolved. Since the authors have decided to claim for the general application of their sampling method beyond images, I believe additional experiments for natural language tasks in their final copy of the work will be really helpful for improving the completeness of this work.

---

> > > ### Author Response · Authors · 2024-08-11
> > >
> > > Thank you for your feedback and for raising your score. We appreciate your recognition of our method's potential beyond image tasks. We will incorporate these additional experiments carefully in the final version to demonstrate the broader applicability of our method.

---

### Author Rebuttal · Authors · 2024-08-06

Dear Reviewers and Area Chair,

Thank you for your thoughtful and constructive feedback on our submission. We greatly appreciate the time and effort you have taken to review our work. We have carefully considered each of your comments and would like to address the all of points raised by the reviewers.

Before addressing specific comments from each reviewer, we would like to address some general or common concerns that were raised:

**Question 1: Theoretical analysis of why the proposed low-dimensional representation of samplers works well.**

We agree that a discussion on the theoretical bounds or guarantees would strengthen our paper. We will expand our manuscript to include a theoretical analysis section focusing on the bounds of the representation error of SS algorithm.

The true sampling function $\tau(x)$ is defined as:
\begin{equation}
       \tau(x) = F(\boldsymbol{f}_1(x), \boldsymbol{f}_2(x), \ldots, \boldsymbol{f}_N(x))
\end{equation}
where $\boldsymbol{f}_i(x)$ are the features of instance $x$.

The approximation $\hat{\tau}(x)$ is defined as:

   \begin{equation}
   \hat{\tau}(x) = H(T(G(x)))
   \end{equation}
   where $G(x) = \sum_{i=1}^{N} \boldsymbol{c}_i \cdot \boldsymbol{f}_i(x)$.

Assuming $F$ is Lipschitz continuous with constant $L$:
   \begin{equation}
   |F(\boldsymbol{f}(x)) - F(\boldsymbol{f}(y))| \leq L \cdot \|\boldsymbol{f}(x) - \boldsymbol{f}(y)\|_2
   \end{equation}

The representation error $\epsilon(x)$ is bounded by:
   \begin{equation}
   \epsilon(x) = |F(\boldsymbol{f}(x)) - H(T(G(x)))|
   \end{equation}
   \begin{equation}
   \epsilon(x) \leq |F(\boldsymbol{f}(x)) - F(\boldsymbol{f}(y))| + |F(\boldsymbol{f}(y)) - H(T(G(x)))|
   \end{equation}
   \begin{equation}
   \epsilon(x) \leq L \cdot \|\boldsymbol{f}(x) - \boldsymbol{f}(y)\|_2 + |F(\boldsymbol{f}(y)) - H(T(G(x)))|
   \end{equation}
   Assuming $\boldsymbol{f}(y) = \hat{\boldsymbol{f}}(x)$:
   \begin{equation}
   \epsilon(x) \leq L \cdot \|\boldsymbol{f}(x) - \hat{\boldsymbol{f}}(x)\|_2 + |F(\hat{\boldsymbol{f}}(x)) - H(T(G(x)))|
   \end{equation}
   Since $H(T(G(x)))$ approximates $F(\hat{\boldsymbol{f}}(x))$:
   \begin{equation}
   \epsilon(x) \leq L \cdot \|\boldsymbol{f}(x) - \hat{\boldsymbol{f}}(x)\|_2 + \epsilon'
   \end{equation}

This bound indicates that the error introduced by our low-dimensional representation is controlled by the Lipschitz constant of the sampling function $F$ and the error in the feature space representation, plus a small approximation error $\epsilon'$.

---

**Question 2: The paper uses a fixed set of features (loss and renormalized entropy) for experiments. It would be helpful to explore how the choice of features impacts the performance of the method or whether different tasks might benefit from different feature sets. This leaves open questions about the flexibility and adaptability of the approach.**

To address your concerns, we have conducted additional experiments to analyze the effect of different feature sets on the performance of our method. We expanded our experiments to include a broader range of features, such as gradient norm and prediction confidence, alongside the original features (loss and renormalized entropy).

We conducted experiments on CIFAR-10 and CIFAR-100 datasets using the following feature sets:

1. **Original Feature Set:** Loss and Renormalized Entropy
2. **Extended Feature Set 1:** Loss, Renormalized Entropy, and Gradient Norm
3. **Extended Feature Set 2:** Loss, Renormalized Entropy, Gradient Norm, and Prediction Confidence

As shown in **Table 9** of the one-page PDF, the results show that the original feature set (Loss and Renormalized Entropy) provides the best performance on both CIFAR-10 and CIFAR-100 datasets, even when compared to extended feature sets including gradient norm and prediction confidence. While gradient norm and prediction confidence can provide additional information, they often overlap with the information captured by loss and renormalized entropy. This redundancy can dilute the effectiveness of the feature set, as evidenced by the marginal improvements or even slight declines in performance observed with the extended feature sets.

---

**Question 3: Lack of discussion on potential computational overhead introduced by the Swift Sampler method. The additional cost of feature computation, Bayesian optimization, and fine-tuning steps could be significant, especially for large datasets.**

We conducted additional experiments to measure the relative training time for each sampling method used in the **Table 4** of the one-page PDF.

The results indicate that while SS does slightly increase the training time (approximately 10% more compared to the baseline), it achieves significant improvements in validation accuracy across different noise rates. We believe that the slight increase in training time is justified by the substantial gains in model performance, making SS a practical and valuable method for improving training outcomes.

---

Once again, we thank the reviewers and the area chair for their valuable suggestions. We have made every effort to incorporate these changes into the paper, and we believe the revised version is more comprehensive and accurate. We hope that our responses and revisions adequately address your concerns and demonstrate the value of our work.

If you have any further questions or require additional clarification, please do not hesitate to contact us during the discussion period.

---

### Decision · Program_Chairs · 2024-09-25

**Decision:**

Accept (poster)

**Comment:**

The paper proposes a novel data sampling approach for supervised training on a fully labeled dataset, to improve training convergence and model performance. It utilizes low-dimensional representations of sampling strategies and Bayesian Optimization for the same. \
The paper received five reviews. The reviewers carefully engaged with the paper and participated in subsequent discussions. The reviews found the paper to be readable and the quality of writing to be good. They found the paper to be well motivated, addressing an important problem, of high relevance to the AI/ ML community. The approach was found to be compelling and novel, reasonably combining ideas from multiple areas to construct the solution. The results demonstrated consistent performance improvement across a range of datasets, tasks, and model architectures, including large-scale problems, suggesting generalizability and broad applicability. \
The initial reviews were mixed, with concerns ranging from (a) lack of theoretical analysis, (b) generalizability across domain (images), tasks (classification), datasets (very small or very large, severe class imbalance), dependence on optimizers and architecture families, (c) rigor of evaluation – proper validation, ablation studies for components, sensitivity analysis for hyperparameters etc., and (d) computational overhead. \
The authors have addressed these concerns in detail, to the satisfaction of reviewers. This has resulted in an upward revision of scores by several reviewers. An updated version of the paper incorporating the authors’ responses to reviewer suggestions and queries (in the main paper/ supplementary) will result in a very strong submission to the conference.